# Bioreplicated coatings for photovoltaic solar panels nearly eliminate light pollution that harms polarotactic insects

Benjamin Fritz[1☉], Gábor Horváth[2☉]*, Ruben Hünig[3], Ádám Pereszlényi[2], Ádám Egri[4,5], Markus Guttmann[6], Marc Schneider[6], Uli Lemmer[1,6], György Kriska[4,7], Guillaume Gomard[1,6]

1 Light Technology Institute (LTI), Karlsruhe Institute of Technology (KIT), Karlsruhe, Germany,
2 Environmental Optics Laboratory, Department of Biological Physics, ELTE Eötvös Loránd University, Budapest, Hungary, 3 Center for Solar Energy and Hydrogen Research Baden-Württemberg (ZSW), Stuttgart, Germany, 4 MTA Centre for Ecological Research, Danube Research Institute, Budapest, Hungary, 5 MTA Centre for Ecological Research, Evolutionary Systems Research Group, Tihany, Hungary, 6 Institute of Microstructure Technology (IMT), Karlsruhe Institute of Technology (KIT), Eggenstein-Leopoldshafen, Germany, 7 Group for Methodology in Biology Teaching, Biological Institute, ELTE Eötvös Loránd University, Budapest, Hungary

☉ These authors contributed equally to this work.
* gh@arago.elte.hu

**Data Availability Statement:** All relevant data are within the paper and its Supporting Information files.

## Abstract

Many insect species rely on the polarization properties of object-reflected light for vital tasks like water or host detection. Unfortunately, typical glass-encapsulated photovoltaic modules, which are expected to cover increasingly large surfaces in the coming years, inadvertently attract various species of water-seeking aquatic insects by the horizontally polarized light they reflect. Such polarized light pollution can be extremely harmful to the entomofauna if polarotactic aquatic insects are trapped by this attractive light signal and perish before reproduction, or if they lay their eggs in unsuitable locations. Textured photovoltaic cover layers are usually engineered to maximize sunlight-harvesting, without taking into consideration their impact on polarized light pollution. The goal of the present study is therefore to experimentally and computationally assess the influence of the cover layer topography on polarized light pollution. By conducting field experiments with polarotactic horseflies (Diptera: Tabanidae) and a mayfly species (Ephemeroptera: *Ephemera danica*), we demonstrate that bioreplicated cover layers (here obtained by directly copying the surface microtexture of rose petals) were almost unattractive to these species, which is indicative of reduced polarized light pollution. Relative to a planar cover layer, we find that, for the examined aquatic species, the bioreplicated texture can greatly reduce the numbers of landings. This observation is further analyzed and explained by means of imaging polarimetry and ray-tracing simulations. The results pave the way to novel photovoltaic cover layers, the interface of which can be designed to improve sunlight conversion efficiency while minimizing their detrimental influence on the ecology and conservation of polarotactic aquatic insects.

**Funding:** This work was supported by the grant NKFIH K-123930 received by Gábor Horváth from the Hungarian National Research, Development and Innovation Office. Ádám Egri was supported by the Hungarian Economic Development and Innovation Operational Programme (GINOP-2.3.2-15-2016-00057), the Hungarian National Research, Development and Innovation Office (grant PD_19-131738) and the János Bolyai Research Scholarship of the Hungarian Academy of Sciences. Benjamin Fritz acknowledges the support of the Karlsruhe School of Optics and Photonics (KSOP, www.ksop.idschools.kit.edu). Furthermore, this study was supported by the German Federal Ministry for Economic Affairs and Energy (BMWi) under contract number 0324179 (CISHiTec).

**Competing interests:** The authors have declared that no competing interests exist.

**Abbreviations:** CIGS, copper indium gallium diselenide, $Cu(In,Ga)Se_2$; GRP, glass-covered rose petal; PMMA, polymethylmethacrylate; RP, rose petal; SBP, smooth black plastic; Brewster's angle, $\arctan(n)$ from the normal vector of the reflecting surface, where $n$ is the refractive index of the reflector's material.

## Introduction

A properly textured front surface of photovoltaic solar panels should allow the following characteristics: (i) A low sunlight reflectance irrespective of the illumination conditions and a high absorption of the collected light in the photovoltaic active layer, both leading to a high energy yield [1–3]. (ii) Radiative cooling that improves the power conversion efficiency and the reliability of the solar panels [4, 5]. (iii) Self-cleaning, which decreases the maintenance costs associated to the removal of soiling particles [6–8]. Although many different multifunctional cover layers have been developed [9–12], their impact on insect ecology and conservation is largely unexplored. The study of this impact is important due to the global insect crisis [13–17] and to the worldwide deployment of photovoltaic installations [18, 19]. These facts urge to understand how the reflection-polarization characteristics of photovoltaic-covered habitats can affect the behaviour of polarization-sensitive insects, especially water-seeking polarotactic aquatic insects being maladaptively attracted to asphalt roads [20] or dark car paints [21], for example.

Many insect species use polarization of light reflected from natural or artificial terrestrial surfaces for object identification or water detection [22–28]. From the Brewster's angle, water surfaces reflect light with typical degrees of linear polarization $15\% \leq d \leq 90\%$ and angles of polarization $-10° \leq \alpha \leq +10°$ relative to the horizontal direction, depending on their brightness/darkness, colour and surface roughness [20–23, 29–33]. Water surfaces usually reflect nearly horizontally polarized light [33]. Aquatic insects in general (belonging to the following orders with aquatic or semiaquatic species: Coleoptera, Collembola, Diptera, Ephemeroptera, Hemiptera, Hymenoptera, Lepidoptera, Mecoptera, Megaloptera, Neuroptera, Odonata, Plecoptera, Trichoptera) have therefore evolved to identify water bodies by the horizontal polarization of water-reflected light [22, 23]. This strategy can result in a maladaptive attraction of polarotactic aquatic insects to smooth artificial surfaces like the glass/plactic covers of solar panels, because these surfaces can reflect a similar polarization pattern as water surfaces [23, 30, 31]. This can result in that aquatic insects are unable to escape from the horizontally polarized signal reflected from photovoltaic panels acting as polarization traps [34] (Fig 1A). If the egg-batches of these insects (e.g. mayflies, dragonflies, stone flies, caddis flies) are laid onto photovoltaic modules, they irremediably perish because of dehydration [23]. Since the larvae of these insects develop in water/mud for a few months/years, hydration by dew or rain drops on the solar panels cannot ensure the survival of eggs. This effect is harmful for the aquatic insect populations concerned, and therefore is called polarized light pollution [30].

So far, the reduction of polarized light pollution of photovoltaic panels has been realized in two ways: i) By painting a grid pattern of narrow (1–2 mm width) white lines on the panel surface [34, 35]. This is based on the unattractiveness of striped or spotted animal coats to polarotactic insects [24–26]. ii) with a nanoporous antireflective layer on the glass cover to decrease the degree of linear polarization of reflected light [31]. Method i) can strongly reduce polarized light pollution, however, the light-collecting area then decreases by 1–5%, depending on the density and width of the white lines [34]. Method ii) does not have this disadvantage.

If, for viewing directions near the Brewster's angle the degree of polarization of reflected light is lower than the threshold of polarization sensitivity and/or if the direction of polarization of reflected light deviates from the horizontal by more than a given threshold value (both thresholds depend on the species considered), a polarotactic insect does not get attracted by the surface-reflected polarized light [29]. Thus, the use of appropriately fine-textured photovoltaic cover layers can reduce the maladaptive attractiveness, and thus polarized light pollution, by decreasing the degree of polarization and changing the angle of polarization of reflected light.

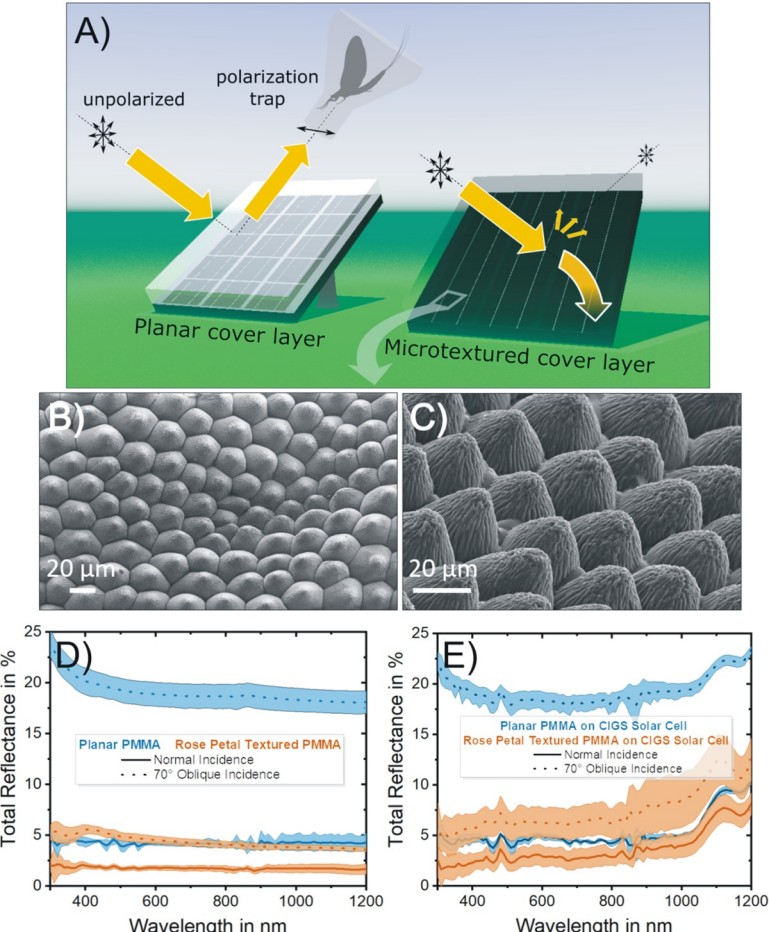

**Fig 1. Optical characteristics of photovoltaic solar panels.** A) Dark photovoltaic modules coated by a reflecting planar cover layer act as polarization traps for polarotactic insects (left) if the photovoltaic-reflected light is partially or completely horizontally polarized. An appropriate texturing of the cover layer strongly reduces polarized light pollution and improves sunlight-harvesting (right). B-C) Scanning electron microscope images of the rose petal replicated cover layer analyzed herein, and incorporating the microtexture of rose petals into a polymeric material (PMMA). Its measured surface reflectance spectrum is shown for both a blackened rear side (D) and for a Cu(In,Ga) Se$_2$ (CIGS) thin-film solar cell coupled to it (E). An untextured ("planar") cover layer is used as a reference and both normal and highly oblique angle of incidences are considered. The coloured areas surrounding the (solid or dotted) curves indicate the standard deviation over N = 4 individual measurements.

We note, however, that although a commercialized antireflective, nanoporous solar glass reduced polarized light pollution for polarotactic horseflies, its attractiveness to certain mayflies was drastically increased relative to that of a smooth reference layer [31]. The reason for this may be that the reflection-polarization characteristics of the black absorber covered with the nanoporous antireflective glass mimicked a ripple-free, smooth water surface for mayflies which prefer calm water bodies to lay their eggs, where their larvae can develop safely [27, 31]. Considering insect conservation, such an approach might therefore cause even more harm to certain insect species than a smooth glass cover.

In this work we demonstrate that microtextured photovoltaic cover layers can strongly reduce their attractiveness to the mayfly species *Ephemera danica* (Müller, 1764) and horseflies (Diptera: Tabanidae), two typical polarotactic aquatic insect taxa [29] functioning as indicators of polarized light pollution. We performed field experiments with microtextured polymeric

coatings incorporating the surface texture of rose petals. These bioreplicated layers improve both light and water management in photovoltaic devices [9, 11, 12]. We show here that they also strongly reduce the attractiveness to mayflies and horseflies. This observation is explained on the basis of their reflection-polarization properties measured with imaging polarimetry, and of raytracing simulations used to analyze the influence of the surface texture topography on the generation of horizontally polarized reflected light.

## Results and discussion

### Topographical quality and light-harvesting properties of polymeric rose petal replicas

Polymeric cover layers replicating the surface texture of rose petals, previously developed for improving sunlight-harvesting in solar cells [11, 12], were fabricated over large areas to experimentally assess their impact on polarized light pollution in the field. The scanning electron microscope images in Fig 1B and 1C show the topography of these replicas. These hot-embossed layers exhibit high structural fidelity with respect to their original bio-template, and replicate both the densely packed array of (epidermal cell) microcones with their mean aspect ratio = 0.6 [9] as well as the (cuticular) nano wrinkles atop, which are kept intact over the double replication process. Their outstanding light-harvesting capabilities are highlighted in Fig 1D and 1E. The rose petal texturing leads to a broadband and angle-tolerant decrease of surface reflectance compared with planar cover layers. For increasing angle of incidence, its optical benefit becomes even more pronounced. Thus, at an angle of incidence = 70˚, the overall reflectance of CIGS solar cells covered with rose petal textured layers can be kept under 10%, a value comparable to the reflectance of planar configurations under normal incidence (see Fig 1E). Overall, the measured reflectance spectra highlight the potential of hot-embossed rose petal replicas as photovoltaic light-harvesting layers.

### Reflection-polarization characteristics

The degree and angle of polarization patterns of (i) an uncoated 10 cm × 10 cm CIGS solar module as well as the same device coated with (ii) a rose petal replica, (iii) a planar PMMA layer and (iv) a microlens array foil were measured with imaging polarimetry for several distinct observation directions under direct sunlight illumination on a cloudless day. The tilt angle of the polarimeter's optical axis was -35˚ from the horizontal, which is near to Brewster's angle $\theta_B \approx 34$˚ from the horizontal for the refractive index $n = 1.49$ of PMMA. The degree and angle of polarization for the four characteristic azimuthal viewing directions of the sun shining from in front of the observer, from behind the observer, from the left and from the right are displayed in Fig 2. In the case of a planar PMMA layer, horizontally polarized light is reflected independent of the observer's azimuthal viewing direction, whereas the rose petal replica only reflects partially horizontally polarized light when observed with the sun shining from in front of the observer. When the sun is shining from behind the observer, a vertical polarization with a vanishingly small degree of polarization is observed. In this case the angle of polarization is surely irrelevant, because the degree of polarization likely falls below the polarization sensitivity threshold of polarotactic insects ([29]), furthermore it may also fall within the uncertainty range of the sensor noise of our imaging polarimeter. Sun shining from the left or from the right results in (opposite) diagonal polarization directions. This distinct reflection-polarization pattern caused by the rose petal replica is qualitatively reproduced by a densely packed, smooth and (close to) hexagonally arranged microlens array cover layer (topography depicted in Fig 3). The uncoated CIGS module, which exhibits surface roughness on the subwavelength scale,

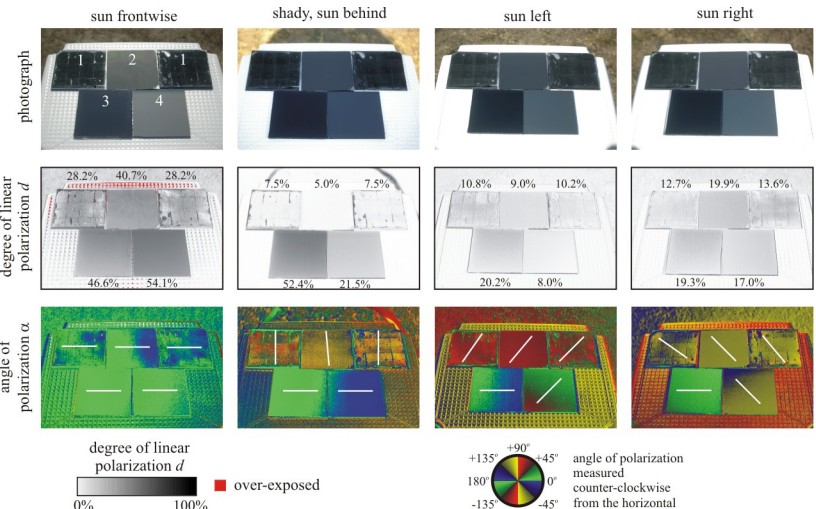

**Fig 2. Imaging polarimetry of the test surfaces.** Polarization patterns of 10 cm × 10 cm CIGS solar modules equipped with various cover layers under clear sky and for different observer viewing directions (columns): 1) Rose petal, 2) artificial microlens array, 3) planar PMMA, 4) no coating. For the cases of sun shining from in front of the observer, from behind the observer, from the left and from the right, three types of images are displayed: Colour photograph (top row), as well as degree of linear polarization d (middle row) and angle of polarization α (bottom row) at 450 nm (blue). In the middle row, the numerical values are the degrees of polarization averaged for the different test surfaces. The white bars in the bottom row show the local average direction of polarization. The tilt angle of the polarimeter's optical axis was set to -35˚ from the horizontal.

also produces the diagonal polarization directions found for the rose petal replica for the sun shining from left and right, but if the sun shines from behind, the uncoated surface reflects horizontally polarized light. We note that the trends in degree and angle of polarization behaviour observed for the uncoated CIGS module are identical to what has been reported for a nanoporous antireflective solar glass [31]. According to Figs 2 and 4 and S1 Fig, both the degree and angle of polarization patterns of all three studied test surfaces depend strongly on

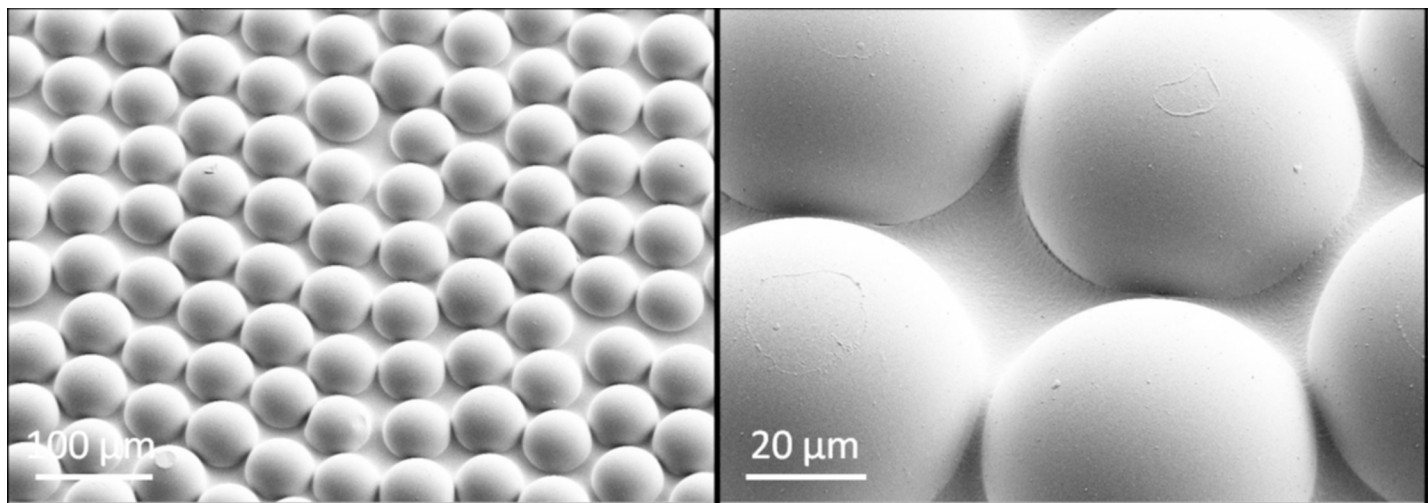

**Fig 3. SEM images of the commercial microlens array foils.** We remind the reader that this is the artificial microlens array (MLA) foil that was included in the measurements of reflection-polarization characteristics of the planar and rose petal textured (PMMA) layers (see Fig 2).

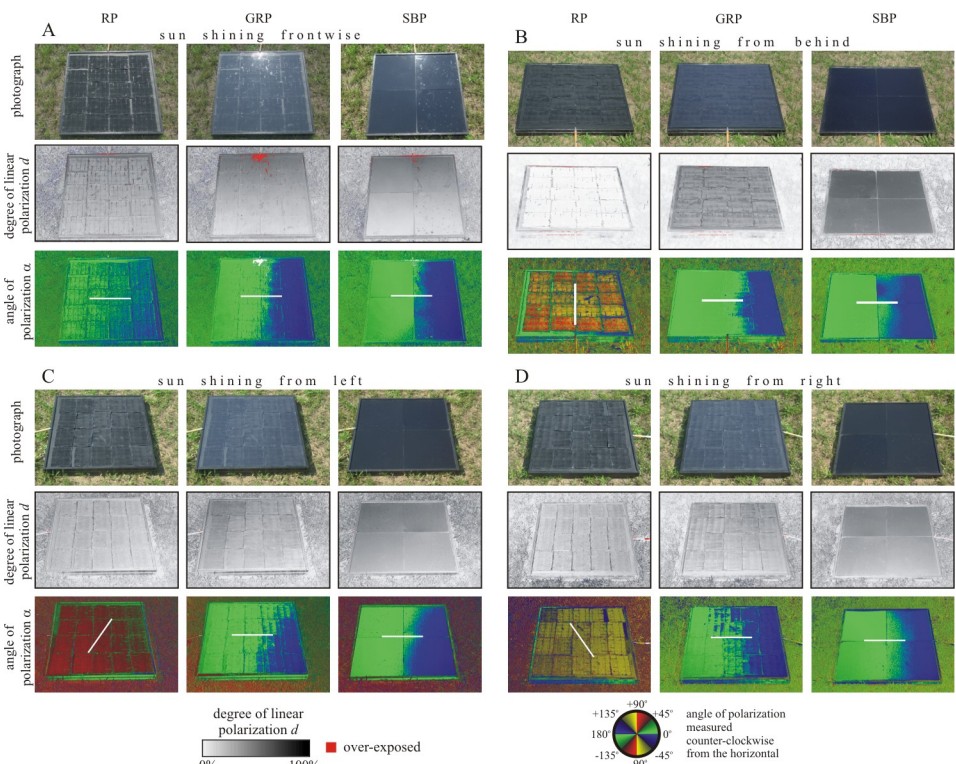

**Fig 4. Photographs of the three test surfaces (RP: Rose petal, GRP: Glass-covered rose petal, SBP: Smooth black plastic) used in the field experiments with horseflies, and corresponding patterns of the degree and angle of polarization.** These patterns were measured with imaging polarimetry in the blue (450 nm) spectral range when the sun shone A) from in front of the observer, B) from behind the observer, C) from the left and D) from the right, as light from the clear sky was reflected from the test surfaces. The tilt angle of the optical axis was -35° from the horizontal. In the angle of polarization patterns, the white bars show the average directions of polarization of the test surfaces.

the observer's viewing direction relative to the position of the sun. A more detailed physical explanation of this ray-optical phenomenon is beyond the scope of this work.

All the reflection-polarization characteristics described in this section are independent of the rose petal replica's orientation, meaning that the reflection-polarization pattern of the rose petal replica remains unaltered if the sample is rotated around its surface normal (see S1 Fig).

Fig 4 shows the patterns of the degree and angle of polarization (at 450 nm, blue) of the three test surfaces used in the field experiments with horseflies when direct sunlight was reflected off the surfaces. By comparison with Fig 2, it can be concluded that if our layers are either applied on a black solar module or on a black absorber layer mimicking such a module, they lead to very similar reflection-polarization properties.

As illustrated in Fig 5A, in the field experiments with mayflies the three test surfaces were placed in close vicinity of vegetation and therefore were not hit by direct sunlight. When blue skylight was reflected off the surfaces (Fig 6A), the rose petal was the least polarizing (degree of polarization $d = 40.2 \pm 16.4\%$, corresponding to the average value ± standard deviation, further on in this work), while the other two surfaces were highly polarizing with practically the same degrees of polarization: $d = 93.9 \pm 3.1\%$ for the glass-covered rose petal and $d = 94.6 \pm 5.5\%$ for the smooth black plastic. Under these conditions, all test surfaces homogeneously reflected horizontally polarized light. When reflected light originating from the leaves of forest

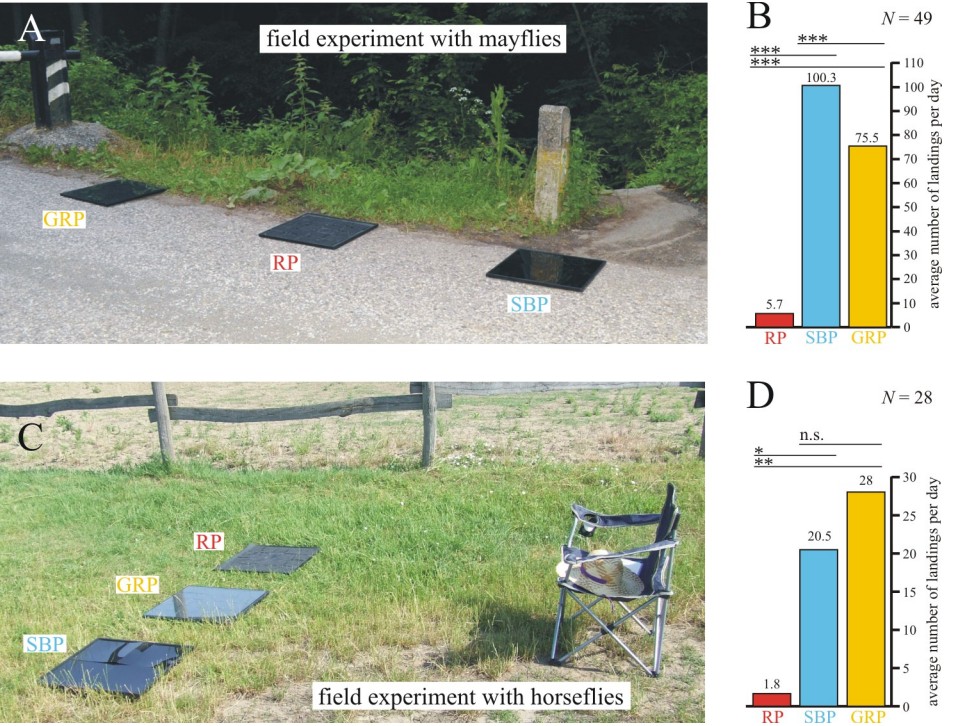

**Fig 5. Arrangements and results of the field experiments with mayflies and horseflies.** The experimental site of the field experiments with mayflies (A) and horseflies (C), including the arrangement of the three test surfaces, and the average daily numbers of landings (B, D) with the results of the Wilcoxon signed-rank test. RP: rose petal, GRP: glass-covered rose petal, SBP: smooth black plastic, $N$: number of observations per category, n.s.: not significant with $p > 0.05$, *: $0.001 < p < 0.05$, **: $0.0001 < p < 0.001$, ***: $p < 0.0001$.

vegetation illuminated by the setting sun was reflected off the surfaces, they were much less and practically similarly polarizing ($16.4\% < d < 21.5\%$) with an approximately vertical direction of polarization (Fig 6B). In Fig 6B the reason for the nearly vertical direction of polarization of reflected light in spite of the horizontal alignment of the test surfaces was the anisotropic illumination: In Fig 6A and 6B the background was the clear sky and forest canopy, respectively. The intensity of light coming from the forest vegetation was much smaller than that of skylight. Consequently, the dominant sideward illumination of the test surfaces in Fig 6B resulted in vertically and weakly polarized reflected light.

The differences between the degrees of polarization of test surfaces used in the experiments with horseflies (Fig 4) and mayflies (Fig 6) are quite striking, because mayflies were studied near sunset under twilight, while horseflies were investigated in full sunshine. At sunset the skylight and light reflected from the surrounding vegetation were the only light sources, while in sunshine the direct sunlight dominated. Thus, the reflection-polarization characteristics of the test surfaces are the consequence of complex interactions between the surface properties and the natural illumination conditions.

To conclude this section, the rose petal replicated light-harvesting layer may cause low polarized light pollution in the field, because: (i) It reflects non-horizontally polarized light for most illumination conditions and directions of observation, making this textured layer clearly distinguishable from water, especially if the observer is in motion. (ii) Even under conditions where the reflected light signal from the rose petal textured layer is horizontally polarized and therefore could be wrongfully detected as water, the lower degree of polarization of reflected

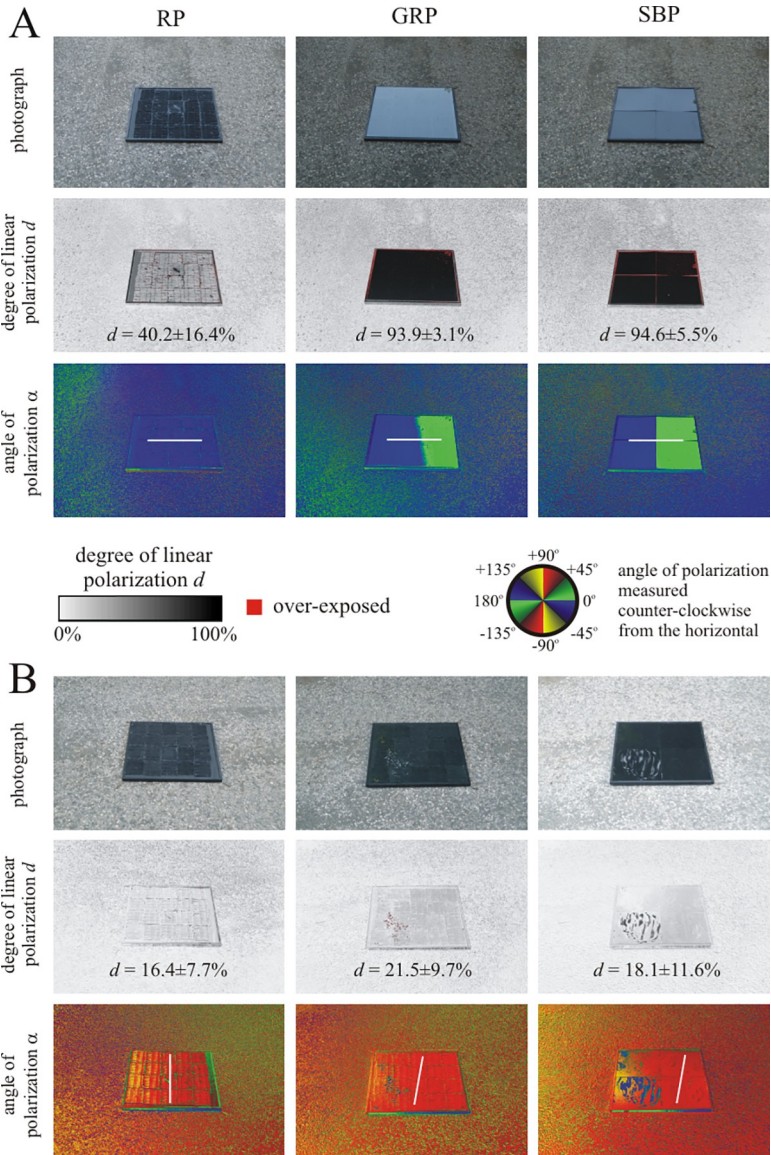

**Fig 6. Photographs and patterns of the degree and angle of polarization of the three test surfaces (RP: Rose petal, GRP: Glass-covered rose petal, SBP: Smooth black plastic) laid on a dry asphalt road in the field experiments with mayflies measured with imaging polarimetry at 450 nm (blue).** In A), the polarimeter's optical axis pointed toward East, approximately parallel to the antisolar meridian, when light from the clear sky was reflected off the test surfaces. In B), the polarimeter's optical axis pointed toward South, approximately perpendicular to the antisolar meridian, when light from the forest vegetation was reflected off the test surfaces. The tilt angle of the optical axis in both cases was -35° from the horizontal. The $d$-values are given in the degree of polarization patterns averaged for the whole test surface. In the angle of polarization patterns, the white bars show the average directions of polarization of the test surfaces.

light may cause considerably smaller visual attractiveness than more efficiently polarizing planar (smooth) configurations. Considering the similarities in degree and angle of polarization patterns when comparing the configurations with black paint at the rearside to the ones with a solar module coupled to the back (Figs 2, 4 and 6, S1 Fig), it is justified to carry out field tests concerning the attractiveness to polarotactic insects by deploying the configuration with blackened rearside.

## Attractiveness of test surfaces to mayflies

In a pilot experiment we used only the rose-petal-replica matte black test surface, which however, did not attract any mayflies due to its low polarizing ability (see Fig 6). Therefore, we continued our field experiment with the use of two additional control surfaces that polarize the reflected light much stronger. From the pilot experiment we conclude that the rose petal replica reduces significantly the polarized light pollution for mayflies, even if it stands alone.

On each experimental day between 18:30 and 19:15, mainly male *Ephemera danica* mayflies were observed around the test surfaces. Then, in the second phase of swarming, the majority of landing specimens were females often with visible egg-batches. Sometimes they laid their egg-batches on the test surfaces. S1 Table contains the time-resolved number of landings of *Ephemera danica* on the three different test surfaces.

Fig 5B shows the numbers of mayfly landings, the statistical analysis of which was performed for 14 min intervals ($N = 49$). The smooth black plastic (SBP) was the most attractive to mayflies, the attractiveness of the glass-covered rose petal (GRP) was weaker, and the rose petal (RP) was practically unattractive to *Ephemera danica*. The Friedman test was highly significant (p < 0.0001, Friedman chi$^2$ = 116.97, df = 3, Kendall's W = 0.6045). According to the Wilcoxon signed-rank test, the mayfly attractiveness of these three test surfaces differed highly significantly (RP versus SBP: p < 0.0001, RP versus GRP: p < 0.0001, SBP versus GRP: p < 0.0001, $N = 49$). The main reason for these findings can be explained by the facts that (i) *Ephemera danica* detects water with the horizontal polarization of water-reflected light [20, 32], (ii) the higher the degree of horizontal polarization of surface-reflected light, the more attractive the respective surface is to polarotactic mayflies [20, 29], and (iii) the degree of horizontal polarization of light reflected from our test surfaces decreased in the order: SBP > GRP > RP (see Fig 6). Egg laying by mayflies was observed only on smooth black plastic and glass-covered rose petal, but not on rose petal.

## Attractiveness of test surfaces to horseflies

Similarly to the experiment with mayflies, in the horsefly experiment we first presented only the rose-petal-replica matte black test surface, which however, did not attract any horsefly due to its low polarizing ability and because in sunshine it reflected not always horizontally polarized light (Figs 2, 4, S1 Fig). Thus, the alone-standing rose petal replica reduces significantly the polarized light pollution for horseflies, too. Therefore, we added two more polarizing control surfaces in the continuation of the horsefly experiment.

Fig 5D shows the numbers of horsefly landings (detailed in S2 Table), the statistical analysis of which was performed for 29 min intervals ($N = 28$). As was the case for mayflies, the rose petal was again found to be practically unattractive. Consequently, one can conclude that the rose petal replicated photovoltaic light-harvesting layer greatly reduced attractiveness of the two representative indicator species (mayflies and horseflies) tested, and therefore, exerts no significant polarized light pollution. This unattractiveness was caused by the reflection-polarization characteristics of the rose petal replicated test surface, which are distinctly different from those of a planar top layer. The differences in degree of polarization are rather small, therefore differences in the attractiveness may be driven mainly by the angle of polarization. We emphasize, however, that the reactions of polarization-sensitive insects are affected by both degree and angle of polarization. Whereas a planar interface produces exclusively (partial) horizontal polarization in reflected sky/sunlight, the rose petal replica only exhibits horizontal polarization if viewed with the sun shining from in front of the observer (see Figs 2 and 4). Since an approaching insect will always observe a targeted surface from multiple viewing angles and can track the change in polarization direction during its flight, a surface that only

produces horizontal polarization for a narrow range of viewing directions is unlikely to be confused as a body of water [22, 23].

The Friedman test was again highly significant (p < 0.0001, Friedman chi$^2$ = 18.489, df = 2, Kendall's W = 0.3796). According to the Wilcoxon signed-rank test, the horsefly attractiveness of the rose petal (RP) was significantly smaller than that of the glass-covered rose petal (GRP) and smooth black plastic (SBP), but the attractiveness of GRP and SBP did not differ significantly (RP versus SBP: p = 0.0035, RP versus GRP: p = 0.00028, SBP versus GRP: p = 1, N = 28). In our field experiments we used the glass-covered rose petal as an important control test surface for the bare rose petal. The former has the same substrate as the latter but with a smooth and thus strongly polarizing glass covering. The reflective (glass-covered) treatments tend to have a higher intensity, yet still attracted more insects (given that horseflies in particular tend to be attracted by darker objects), implying that polarization and not intensity was responsible for the different reactions of the studied polarotactic insects.

To the best of our knowledge, the polarized light pollution caused by a photovoltaic light-harvesting layer has so far only been investigated for the case of antireflective solar glass incorporating nanopores [31]. Although the attractiveness to horseflies could thereby be reduced, mayflies were actually significantly more attracted to the solar glass compared to a planar reference cover layer. We note that the respective experimental site was identical to our field studies with mayflies. The microtextured layer investigated herein therefore exhibits the novel property of minimizing polarized light pollution at least for the studied mayflies and horseflies. Comparing the degree and angle of polarization patterns, we suspect that the reason for this broadband applicability compared to the antireflective layer studied in reference [31] can be attributed to the fact that the rose petal microtexture reduces the degree of polarization of reflected light with much greater extents than the nanoporous antireflective solar glass, depending on the angle of reflection. Thus, the rose petal treatment was much less attractive to water-seeking mayflies and water- or host-seeking horseflies under the studied illumination conditions.

The horsefly attractiveness of glass-covered rose petal did not differ significantly from that of smooth black plastic. On the other hand, the mayfly attractiveness of glass-covered rose petal was significantly smaller than that of smooth black plastic. The reasons for this attractiveness difference may be the different illumination conditions in the field experiments and the species-dependent reactions of horseflies and mayflies to these two different black planar test surfaces.

## Numerical assessment of reflection-polarization patterns of microtextured surfaces

It was experimentally shown that the disordered microtexture replicated from rose petals greatly reduced polarized light pollution. By comparison of its reflection-polarization characteristics with those of a microlens array foil (see Fig 2), we also concluded that such dense arrays of micron-sized textural elements lead to a distinct pattern in the angle of polarization for different observer viewing directions, irrespective of the exact curvature of the individual building blocks. In what follows, we analyze more systematically the influence of the texture topography on polarized light pollution to derive general guidelines for the design of photovoltaic cover layers. To this end, a ray-tracing based numerical assessment of the polarized light pollution (depending on the angle of incidence) caused by microcone arrays of varying aspect ratio as well as varying degrees of both height and positional disorder (see Fig 7C–7E) were carried out according to the methodology described in section 2.7.

As reported in [29], surfaces reflecting light with a degree of polarization $d > 15\%$ and a polarization direction deviating by maximum ±10° from the horizontal cause a maladaptive

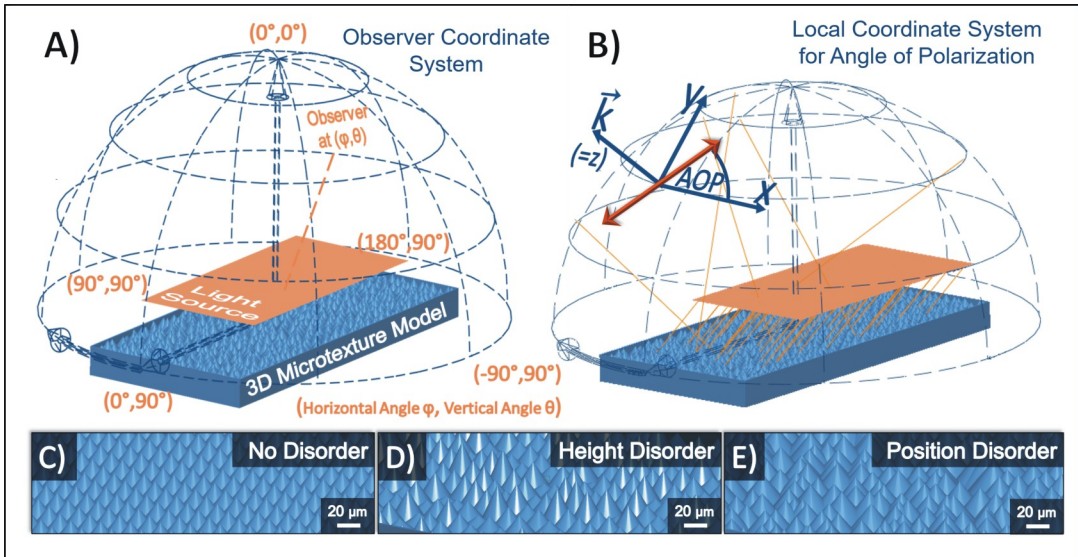

**Fig 7. Illustration of a ray-tracing-based simulation approach for studying the properties of light reflected from densely packed and disordered microtextures.** A) Definition of the spherical coordinate system for characterizing the observer's viewing directions in the farfield. The planar rectangular and transparent light source is marked by an orange rectangle. It emitted $10^8$ parallel and unpolarized rays of identical initial power with random starting positions into the central part of the microtexture models. B) At every observer viewing direction ($\varphi$, $\theta$), the incoming rays' (exemplary ray paths are drawn as faint orange lines) propagation direction $\underline{k}$ defines the $z$-axis of a local coordinate system that is used for measuring the angle of polarization (noted AOP in the figure). The $y$-axis is chosen parallel to the local $\varphi$ = constant line. The angle of polarization of the electric field vector of light represented by an orange double-headed arrow is then always measured relative to the local $x$-axis (mathematically positive). The array of parallel orange lines under the orange rectangle represents parallel sun rays arriving from an intermediate elevation near an azimuth of 180˚. C)-E) depict exemplary microtexture models (cone's aspect ratio = 0.6, full tiling of the base) illustrating three extreme cases, namely the disorder-free configuration (C), the maximum degree of height disorder (D) and the maximum disorder in the cones' horizontal position (E).

attractiveness to mayflies and horseflies because of misidentification as a water surface. These thresholds can be exploited to quantify the polarized light pollution (and its dependence on several structural parameters) that is caused by the reflection of direct sunlight from microtextured surfaces. We assume that the textured surface is placed horizontally on the ground so that it can be observed from all positions on the $2\pi$ hemisphere (see Fig 7A and 7B). Furthermore, on the basis of numerous field experiments [e.g. 20, 21, 27–29, 31, 34], we assume that none of the possible observer viewing directions is favoured by the insects for spotting water surfaces. From the farfield distributions of reflected light intensity, degree and angle of polarization, we then calculated three numbers for characterizing the polarized light pollution:

- The ratio of the solid angles $S_{illum}/2\pi$. $S_{illum}$ is the total solid angle within which a nonzero intensity of reflected light was collected. This ratio provides a measure of the spread/concentration of reflected light over the hemisphere of observer viewing directions.

- The ratio of the solid angles $S_{attract}/S_{illum}$. $S_{attract}$ is the total solid angle for which both a degree of polarization $d > 15\%$ and a polarization direction that only deviates by maximum $\pm10\%$ from the horizontal is found. Therefore, $S_{attract}/S_{illum}$ measures which portion of these observer viewing directions can be suspected to cause mayflies and horseflies to mistake a panel for a watery surface.

- The product of these two ratios, $S_{attract}/2\pi$, is the total solid angle from which a misdetection of the surface as water can be suspected relative to the whole hemisphere.

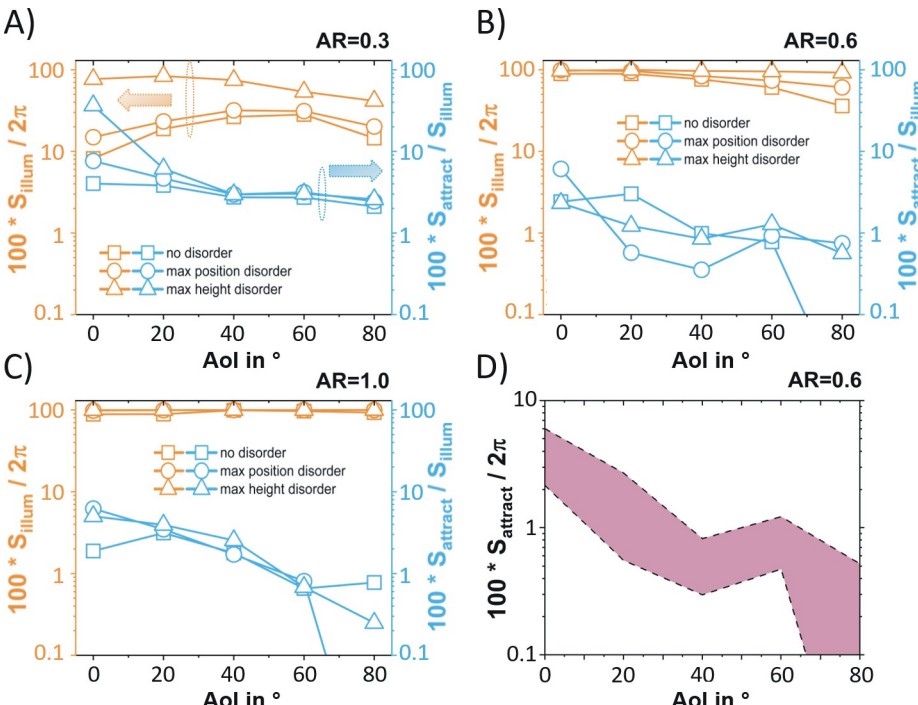

**Fig 8. Numerical assessment of the influence of structural disorder on the polarized light pollution caused by microtextured surfaces.** For aspect ratio AR = 0.3 (A), 0.6 (B), and 1 (C), the two characteristic solid angle ratios $S_{illum}/2\pi$ and $S_{attract}/S_{illum}$ are displayed as a function of the angle of incidence AoI. D) Depicts the solid angle fraction $S_{attract}/2\pi$ versus AoI for which horseflies and mayflies would detect the reflecting surface as a water surface with respect to the entire hemisphere of possible observer viewing directions for aspect ratio AR = 0.6. The shaded area in panel D indicates the full range of numerical results that were found when ramping up both height and position disorder.

With a focus on aspect ratio = 0.3, 0.6 (close to the average aspect ratio of rose petal epidermal cells [9]) and 1, the results of these computations are displayed in Fig 8. We found that for the whole ranges of the aspect ratio and the angle of incidence considered, it can first be concluded that both disorder in cone positioning and disorder in cone height lead to a larger portion of the observer hemisphere being hit by reflected light. This is especially pronounced for disorder in cone height, since the slight variations in local angle of incidence for neighbouring cones leads to a peak broadening effect in the reflected light farfield intensity (see S2–S4 Figs and orange curves in Fig 8A–8C). Except for the cases of (at least close to) perfectly ordered and cones with low aspect ratio, the microtextured surfaces are able to spread the incident parallel light over most of farfield viewing directions (φ, θ). Considering the portion of the hemisphere from which the reflecting surface could be wrongfully detected as water, an analogous statement for the influence of disorder can not be made. However, we note that (with the exception of aspect ratio = 0.3 at angle of incidence = 0°) this portion is smaller than 10%, and even decreases below 1% for high angle of incidences. The calculated fractions of possible farfield viewing directions (φ, θ) that lead to wrongful detection as a water surface relative to the whole hemisphere are summarized in Fig 8D. The shaded area in Fig 8D indicates the full range of numerical results that were found when ramping up both height and position disorder. For the entire ranges of the aspect ratio, angle of incidence and disorder type and amounts that were investigated herein, the conditions for a misdetection as water are only met for a very limited range on the observer hemisphere: according to Fig 8D, across all angle of incidences, less than 10% of the 2π hemisphere (a solid angle withe area smaller than 0.2π) would

fool the insects. This remarkable robustness leads to the conclusion that the ability of micro-textured cover layers, including polymeric petal replicas, to drastically diminish polarized light pollution is not strongly susceptible to changes in illumination conditions as well as to variations of geometrical parameters like the aspect ratio or the type and degree of structural disorder.

For the design of multifunctional microtextured photovoltaic cover layers, the target of causing minimal polarized light pollution therefore only constitutes a weak constraint on the design parameters. With a clear advantage over nanoporous antireflective light-harvesting solar glass layers [31] in terms of reducing polarized light pollution, our microtextured rose petal imitating surfaces therefore seem to be a promising pathway towards, at the same time, achieving outstanding light-harvesting properties [9–12] as well as introducing a self-cleaning scheme [11] and diminishing maladaptive attractiveness to polarotactic aquatic insects.

## Conclusion

Photovoltaic cover layers form the interface between solar modules and their environment. They should ideally maximize sunlight-harvesting while minimizing potential deleterious ecological impact, such as polarized light pollution. In this study, we hence focused on their maladaptive attractiveness to polarotactic insects, which can get trapped by the horizontal polarization of light reflected from photovoltaic modules. Our main objective was to answer the following questions: Can microtextured cover layers, in addition to improving sunlight capture, also serve for limiting polarized light pollution and what are the key topographical features involved? To this end, we analyzed polymeric replicas of the natural rose petal surface texture as an example for a multifunctional photovoltaic cover layer. Field experiments conducted on polarotactic mayflies and horseflies in Hungary demonstrated that the planar reference surfaces attracted many more landings than the microtextured rose petal surface. For mayflies and horseflies, respectively, approximate glass-covered rose petal/rose petal ratios were 14 and 17, and smooth black plastic/rose petal ratios, 25 and 12. Thus the rose petal surface greatly reduced polarized light pollution. Imaging polarimetry further proved that light reflected by these microtextured cover layers is mostly non-horizontally and weakly polarized, independent of illumination and observation conditions. By comparing the measured reflection-polarization characteristics of the rose petal micro-/nanotexture and of an array of identical and smooth microlenses, it was concluded that the conical microstructures were mostly responsible for that effect. Ray optics simulations were additionally performed on an array of smooth microcones with varying aspect ratios and degrees of structural disorder. It was inferred that the analyzed microtextures exhibit a negligible risk of polarized light pollution, even for the different illumination conditions and topographies considered. Summing up, our results indicate that microtextured cover layers can simultaneously improve sunlight-harvesting and limit the detrimental effect of polarized light pollution for polarotactic (aquatic) insects. From an engineering viewpoint, this study shows that the conditions for achieving low degrees of horizontally polarized light are not restrictive, which gives a lot of freedom for the optical design of microtextured photovoltaic cover layers with low polarized light pollution.

## Materials and methods

### Animal ethics statement and field study permits

Csaba Viski permitted us to photograph his horses in his horse farm in Szokolya. For the location and activities of our field study no specific permissions were required.

The main focus of this work is to compare the attractiveness of three types of solar panel surfaces to polarotactic aquatic insects (mayflies and horseflies). Large-area polymeric replicas

of the rose petal surface texture were chosen, because we expected that, due to their very rough surface [9, 36], they reflect sun/skylight so diffusely (i.e. with low degrees of polarization and not always horizontal direction of polarization), that their attractiveness to polarotactic insects may be minimal. As control surfaces, we chose a glass-covered rose petal surface and a smooth black plastic surface (made from the same material as the large-area petal replica). Since, due to their smooth and black surface, the two latter panels polarize reflected light strongly (with high degree of polarization) and horizontally at the Brewster's angle, they were expected to be attractive to polarotactic aquatic insects.

## Fabrication and topographical analysis of large-area rose petal polymeric replicas

The 50 cm × 50 cm rose petal textured test surfaces used to study the attractiveness of different solar panel cover layer configurations to polarotactic insects were assembled from 20 identical, large-area (12.5 cm × 10 cm) rose petal replicas, which were fabricated in poly(methyl methacrylate) (PMMA, upag AG, Germany) by employing an upscaled hot embossing replication routine [11, 12]. In short: several natural rose petals were manually and very carefully assembled into an uninterrupted "array" and fixed onto a flat substrate. After a two-step replication of this large-area rose petal biotemplate via polydimethylsiloxane casting and subsequent soft imprint replication into a polymeric material, a durable and temperature-stable metallic (negative) mold is fabricated from this upscaled replica. This metallic mold is subsequently used for hot embossing self-standing PMMA foils, enabling the reliable and high throughput fabrication of petal texture replicas. A detailed description of this large-area replication routine, the consecutive lamination onto solar modules and a performance assessment can be found in reference [12].

After a continuous outdoor operation of various solar panels equipped with rose petal textured PMMA antireflective layers over a full year in Karlsruhe (Germany), no detectable decrease in structural quality or in opto-electrical performance was found. We further note that PMMA has already been shown to meet the necessary durability standards for photovoltaic applications [37]. However, accelerated ageing tests that show the long-term outdoor durability of such microtextured PMMA antireflective layers on top of solar panels are still pending.

The topography of the resulting large-area rose petal replicas was analyzed using a ZEISS SUPRA 60 VP scanning electron microscope. Furthermore, the scanning electron microscope images in Fig 3 were acquired using a ZEISS SUPRA 55 VP SEM.

## CIGS solar modules with antireflective cover layers

Copper indium gallium diselenide (Cu(In,Ga)Se$_2$, CIGS) thin-film solar modules (only bare layer stack without monolithic wiring) with dimensions of 10 cm × 10 cm were equipped with different cover layers to measure their antireflective properties as well as their reflection-polarization characteristics. All cover layers were laminated onto the CIGS modules using the highly transparent liquid adhesive NOA88 (Norland Products, Inc., USA), which solidifies when exposed to ultraviolet light. The devices were fabricated in a co-evaporation process, as described in [38]. To prepare the samples, only pieces of these modules with dimensions of 2.5 cm × 2.5 cm were used due to the limited size of the integrating sphere used for the optical characterization (see Subsection 2.3.).

## Light-harvesting properties of test surfaces

To assess the light-harvesting capabilities of photovoltaic cover layers, we measured their angle-dependent surface reflectance (specular + diffuse components) using a Lambda 1050

ultraviolet/visible/near infrared spectrophotometer (PerkinElmer, Inc, USA) equipped with an integrating sphere and a pivotable sample holder. In order to minimize the reflection of light from the samples' rear surface, their backsides were painted black (with acrylic Decorlack 073, Marabu GmbH & Co. KG, Germany). To achieve full and homogeneous coverage, at least three layers of paint were applied. Comparing the angle-dependent reflectance of a black-painted PMMA reference sample with planar surface to theoretical values obtained by using Fresnel's equations for an air-PMMA interface proved that the black paint keeps rearside reflection sufficiently low for all angle of incidences and over the whole spectral range relevant for (CIGS) photovoltaic modules, that is 300 nm—1200 nm (not shown here because of lack of space). As well as the bare surface reflectance, their overall reflectance spectra were also acquired for planar and textured PMMA layers laminated (as briefly described in subsection 2.2.) onto CIGS thin-film solar cells.

## Reflection-polarization characteristics of test surfaces

The reflection-polarization properties (degree and angle of polarization) of the test surfaces used in the field as well as those of various photovoltaic cover layers were measured with rotating-analyser sequential (serial) imaging polarimetry [22: Chapter 1, pp. 1–14] in the red (650 nm), green (550 nm) and blue (450 nm) spectral ranges from different directions of view and under different (sunny and overcast) illumination conditions. The tilt angle of the optical axis of the polarimeter pointing toward the target (reflecting test surfaces) was measured from the horizontal so that negative angles mean directions below the horizon, oppositely to the $z$ axis direction in Fig 7B. Since many aquatic insect species are sensitive to reflected polarization at shorter wavelengths [23, 39–41], here we chose to present only the polarization patterns measured at 450 nm, since all of the configurations considered in this study lead to practically identical patterns for the green and red spectral ranges, and our polarimeter functioned only in the visible spectral range. In addition to planar and rose petal textured PMMA layers, an artificial polymeric microlens array foil (Lumlight Product No. MA1303001, Lumtec Lighting Corp., Taiwan) was laminated onto a CIGS module and its reflection-polarization characteristics were measured (see Fig 2). Scanning electron microscope images of the microlens array topography can be found in Fig 3.

Imaging polarimetric measurements were performed with a NIKON D3200 DSLR digital camera (24–70 mm f/2.8) equipped with a rotatable linear polarizer (W-Tianya Slim MCCPL with a circular filter frame of 2 mm thickness) fixed on a tripod to eliminate motion artefacts. The camera's roll axis (coinciding with its optical axis) was levelled to horizontal with a common bubble-tube. In our polarimetric measurements the exposure, sensitivity and aperture of the camera were set manually and other auto adjustments were disabled. The aperture (f-number) was always set to the medium value of 5.6, and in order to reduce sensor noise, the ISO sensitivity was set to a relatively low value of 800. Depending on the intensity of light reflected from the measured test surface, the exposure time changed between 1/60 and 1/500 seconds. To ensure a larger depth of field for the studied horizontal test surfaces, optical vignetting was intentionally not reduced. Pixel vignetting was not compensated, and the mechanical/accessory vignetting was minimal, because there was no lens hood and the circular frame of the linearly polarizing filter was thin (2 mm). We did not use any post-processing software to compensate for the remaining small vignetting effect. The polarization images taken through the polarizer were saved in uncompressed Nikon raw format. Under laboratory conditions we tested that our polarimeter can measure the degree of polarization with an accuracy of ±1%, and the angle of polarization with a precision of ±1˚.

In Figs 2, 4, 6 and S1 Fig, overexposed pixels (> 90% of the maximum pixel byte value 255) are labelled by red, because these may provide inaccurate degree of polarization. Since the

manually set exposure- and aperture-values of the camera of our imaging polarimeter were optimized to measure the reflection-polarization patterns of the (shiny or matte) black test surfaces, in Figs 2, 4, 6 and S1 Fig there are no underexposed pixels (< 10% of the maximum pixel byte value 255).

## Field experiments with mayflies

To test the attractiveness of different photovoltaic cover layers to *Ephemera danica* mayflies, we performed field experiments in the North Hungarian Mountains close to the village of Dömörkapu at a bridge overarching the Bükkös Creek (47˚ 41' 45" North, 18˚ 59' 50" East) between July 6th and July 13th 2019 on 6 warm days (with 25–35˚C average maximum daily temperatures) from 18:30 to 21:00 (= local summer time = Universal Time Coordinated + 2 hours). We deployed three different 50 cm × 50 cm test surfaces (1 m spacing) on an asphalt road parallel to the creek. According to our earlier field experiments with mayflies [20, 21, 29, 31, 32, 34], the 1 m spacing proved optimal to ensure simultaneously the independence and the minimization of site effect of the reactions to neighbouring test surfaces. The experimental site is depicted in Fig 5A. The following three different test surfaces were investigated: (i) A hot-embossed rose petal replica (50 cm × 50 cm) called simply as 'rose petal' further on, (ii) the same polymeric replica covered with a common glass plate of 3 mm thickness called as 'glass-covered rose petal' further on, and (iii) a planar PMMA reference layer composed of four quadratic elements (25 cm × 25 cm) called as 'smooth black plastic' further on. The order of these three test surfaces was changed with cyclic permutation (1-2-3, 2-3-1, 3-1-2, and so on) after every 14 min within 10 seconds, then 1 min was waited such that the mayfly behaviour can be recovered from any disruptive effects of humans moving around the test panels. As for the samples analyzed in Fig 1D, the backsides of the test surfaces were painted black with 3 layers of black acrylic paint (Decorlack 073, Marabu GmbH & Co. KG, Germany) to mimic the appearance of highly absorbing photovoltaic modules. The experimental site was surrounded by trees, thus the surfaces were illuminated by the down-welling skylight and light reflected from the green vegetation.

*Ephemera danica* mayflies arrived from the creek, then flew along the road and met our test surfaces. When they reached the region of the three test panels, they often landed on one or two of them and finally flew away. The number of landing events was counted for each test surface in 14 min intervals. When a mayfly performed more consecutive landings, only the first one was counted. At a given point of time landing(s) of only one mayfly individual occurred, thus the counting of landing events was easy. The number of mayfly landings for the different surface configurations and for every 14 min intervals can be found in S1 Table. For the statistical analysis (Friedman test with Kendall's W effect size and Wilcoxon signed rank test with Bonferroni correction), these 14 min intervals were used.

Water-seeking mayflies are usually attracted to horizontally polarizing asphalt surfaces [20, 22, 23, 29, 30, 32]. In these experiments the three different test surfaces were laid on a dry asphalt road as a background. Since the degree of polarization of asphalt-reflected light was much lower than that of test-surface-reflected light (see Fig 6), furthermore the asphalt surface below and near the test surfaces was homogeneous, the asphalt did not affect the overall attractiveness to polarotactic mayflies. Therefore, the differences in the attractiveness of the three test surfaces were surely not because of the weak and homogeneous reflection-polarization signal of the asphalt road.

## Field experiments with horseflies

Similar to the field experiments with mayflies, a measurement campaign focusing on horseflies was performed in June 2019 on four sunny and warm days (with 25–35˚C average maximum

daily temperatures) on a Hungarian horse farm in Szokolya (47˚ 52' North, 19˚ 00' East). The same three test surfaces as in the field experiments with mayflies (see subsection 2.5.) were placed on the grassy ground along a straight line, 50 cm apart from each other. According to our earlier field experiments with horseflies [21, 24–29, 31, 34], the 50 cm spacing proved to be optimal to ensure simultaneously the independence and the minimization of site effect of the reactions to neighbouring test surfaces. Their order was changed with cyclic permutation (1-2-3, 2-3-1, 3-1-2, and so on) every 29 min within 10 seconds, then 1 min was waited for recovering the normal behaviour of horseflies. They were exposed to direct sun- and skylight and were never in the shadow of the vegetation. The experimental site is depicted in Fig 5H. An observer wearing white clothes counted the number of landings of horseflies from a chair placed at a distance of 2 m from the test surfaces. When a horsefly performed more consecutive landings, only the first one was counted. At a given point of time landing(s) of only one horsefly happened. The total number of horsefly landings for the three different test surfaces and the four experimental days can be found in S2 Table. During this experiment the number of landings was recorded in 29 min intervals. These data were used for Friedman test with Kendall's W effect size and Wilcoxon signed rank test with Bonferroni correction. During this experiment the 4×(6–8) = 24–32 changes of order of the test surfaces were large enough to minimize/eliminate any site/position effect.

Because the field experiments with horseflies were performed by a highly experienced observer of tabanids (G. Horváth), we are confident that the counted flies were all horseflies. Since they were not captured, however, horseflies that landed on test surfaces could not be identified to the species level. Thus, unlike for the mayflies, we could not exclude the possibility of pseudoreplication due to repeated visits by some individuals. In previous field experiments at the same location [27, 28], the following horsefly species were found to occur: *Tabanus tergestinus, T. bromius, T. bovinus, T. autumnalis, Atylotus fulvus, A. loewianus, A. rusticus* and *Haematopota italica.*

## Ray-tracing simulations of reflection-polarization characteristics

A plurality of all existing flowering (Angiosperm) plant species exhibits a common basic structural composition of their micro/nanotextured petal surface [42, 43], namely that their (tens of μm sized) epidermal cells form a densely packed array of micro-protrusions which are decorated with a folded wax layer, being of a minor importance for the texture's optical properties [10]. These protrusions are commonly well described by the (simplified) assumption of a conical shape [44]. On this basis, we developed a modelling approach based on Monte Carlo ray-tracing to numerically investigate the polarization properties of sunlight being reflected from petal-like surface textures. Due to the general complexity and the wide range of possible terrestrial illumination conditions (variable direct and diffuse components), we studied only the reflection-polarization properties under clear sky conditions, which are mainly caused by the direct, parallel component of sunlight. For varying angle of incidence, we numerically investigated the properties of parallel light rays being reflected from microtextured surfaces based on densely packed cones for all possible observer positions in the farfield. In our optical computations the cone radius was always fixed at 10 μm. Note however, that since our optical simulations are based on ray optics, the absolute scale of these models is actually irrelevant (as long as all features are much larger than the wavelength of light) and only the relative proportions (like the aspect ratio) are influencing the optical properties.

Although sharply tipped cones do not fully follow the actual morphology of rose petal epidermal cells, our previous experimental and numerical analyses indicated that such idealized cones nevertheless allow to approximate the angle-dependent reflectance behaviour of petal

texture replicas. Our previous simulation efforts on microtextures based on cones, ellipsoids and pyramids (not reported here) further showed that the general conclusions drawn from the simulations herein would not be affected by slight changes in the texture morphology, since all interactions are solely governed by ray-optical effects, which tend to be more robust against slight structural variations (in contrast to wave-optical effects like interference).

The observer positions are defined in spherical coordinates ($\varphi$, $\theta$) (see Fig 7A). At every ($\varphi$, $\theta$), the intensity of the collected light rays as well as the resulting polarization state, meaning the degree and angle of polarization, are observed. The ($\varphi$, $\theta$)-dependent coordinate system we chose for measuring the angle of polarization is illustrated in Fig 7B. At every observer viewing direction ($\varphi$, $\theta$) in the farfield, a (right-handed) local Cartesian coordinate system is defined. Its $z$-axis is defined by the incoming rays' propagation direction $\underline{k}$. The $y$-direction is chosen parallel to the local $\varphi$ = constant line. With these definitions, the direction of polarization always lies in the local $x$-$y$ plane. The angle of polarization at ($\varphi$, $\theta$) is then measured relative to the local $x$-axis (mathematically positive). In the used ray-tracing software (LightTools version 8.7, Synopsys Inc., USA), both degree and angle of polarization can be calculated either directly from Stokes' parameters for each individual ray in combination with their direction of propagation, or indirectly using a locally defined linear polarizer in the farfield. Both approaches were performed for cross-checking the results. However, all the data displayed herein were obtained using the indirect approach. After performing the ray-tracing algorithm, the linear polarizer is rotated for every observer viewing direction ($\varphi$, $\theta$) around the local $z$-axis to obtain the filter positions for which the power collected for this specific observer viewing direction reaches its maximum and minimum. The filter position can then be directly translated into the local angle of polarization, and the degree of polarization $d$ can be calculated from the maximum and minimum power values $P_{\max}$ and $P_{\min}$, that are found at ($\varphi$, $\theta$) when rotating the linear polarizer:

$$d(\varphi, \theta) = \frac{P_{\max}(\varphi, \theta) - P_{\min}(\varphi, \theta)}{P_{\max}(\varphi, \theta) + P_{\min}(\varphi, \theta)}.$$

Example ray-tracing models for the perfectly ordered as well as the maximum amounts of both height and position disorder are depicted in Fig 7C–7E. Further details about all structural parameters investigated herein and the various convergence tests that were performed to determine reasonable spatial dimensions and numbers of individual microcones (to incorporate structural disorder) for the ray-tracing models, as well as a proper number of rays to trace can be found in the S1 File.

## Supporting information

**S1 Fig. Photographs and patterns of the degree and angle of polarization of the rose petal (RP) replica and the planar PMMA (SBP) reference layer used in the field experiments with mayflies and horseflies for four different orientations of the RP.** The polarization patterns were measured with imaging polarimetry in the blue (450 nm) spectral range when the sun shone from behind (top) and left (bottom) and light from the clear sky was reflected from the test surfaces. The tilt angle of the optical axis was -35° from the horizontal. In the middle row, the numerical values are the degrees of polarization averaged for the different test surfaces. In the angle of polarization patterns, the white bars show the average directions of polarization of the test surfaces.
(DOCX)

**S2 Fig. Simulated farfield reflection-polarization characteristics (light intensity shown in colours, and polarization represented by double-headed arrows, the length of which is**

**proportional to the local degree of linear polarization DoLP) as functions of observer position and angle of incidence AoI.** The fully packed cones have an aspect ratio AR = 0.6 with standard deviations $\sigma_h = \sigma_p = 0$, where $\sigma_h$ is the disorder of cone height and $\sigma_p$ is the disorder of cone position.
(DOCX)

**S3 Fig. As S2 Fig for microcones with aspect ratio AR = 0.6, $\sigma_h = 0$ and $\sigma_p = 0.5 \cdot \bar{d}$, where $\bar{d}$ is the average distance between nearest cone neighbours for the unperturbed, hexagonally arranged model.**
(DOCX)

**S4 Fig. As S2 Fig for microcones with aspect ratio AR = 0.6, $\sigma_h = 0.3 \cdot \bar{h}$ and $\sigma_p = 0$, where $\bar{h}$ is the average cone height.**
(DOCX)

**S1 Table. Number of landings of *Ephemera danica* mayflies on the three different test surfaces (RP: Rose petal, GRP: Glass-covered rose petal, SBP: Smooth black plastic) used in the field experiments on 6, 7, 10, 11, 12 and 13 June 2019.** The daily time period (UTC + 2 hours) of the experiment is also given.
(DOCX)

**S2 Table. Numbers of landings of horseflies on the three different test surfaces (RP: Rose petal, GRP: Glass-covered rose petal, SBP: Smooth black plastic) used in the field experiment on 18, 19, 25 and 27 June 2019.** The daily time period (UTC + 2 hours) of the experiment is also given.
(DOCX)

**S1 File. Ray tracing simulations.**
(DOCX)

## Acknowledgments

We gratefully acknowledge the contribution of the Karlsruhe Nano Micro Facility (KNMF, www.knmf.kit.edu) in providing the platform for the mold insert fabrication and petal texture replication at KIT (www.kit.edu). Furthermore, we would like to thank all colleagues from the IMT workshop for their skillful aid in mechanically processing the Ni mold insert. Markus Wissmann's (IMT, KIT) contribution to acquiring the scanning electron microscope images, as well as Martin Held's support by providing the microlens array foil are also gratefully acknowledged. We are grateful to Csaba Viski, who permitted the performance of our field experiment at his horse farm in Szokolya, Hungary. We thank the constructive and valuable comments of three anonymous reviewers on an earlier version of our manuscript.

## Author Contributions

**Conceptualization:** Benjamin Fritz, Gábor Horváth, Guillaume Gomard.

**Data curation:** Benjamin Fritz, Gábor Horváth, Ruben Hünig, Ádám Pereszlényi, Guillaume Gomard.

**Formal analysis:** Benjamin Fritz, Ádám Pereszlényi, Markus Guttmann, Marc Schneider, Uli Lemmer, György Kriska, Guillaume Gomard.

**Funding acquisition:** Gábor Horváth, Guillaume Gomard.

**Investigation:** Benjamin Fritz, Gábor Horváth, Ruben Hünig, Ádám Pereszlényi, Ádám Egri, Markus Guttmann, Marc Schneider, Uli Lemmer, György Kriska, Guillaume Gomard.

**Methodology:** Benjamin Fritz, Gábor Horváth, Ruben Hünig, Ádám Pereszlényi, Ádám Egri, Markus Guttmann, Marc Schneider, Uli Lemmer, György Kriska, Guillaume Gomard.

**Software:** Benjamin Fritz, Markus Guttmann.

**Supervision:** Guillaume Gomard.

**Validation:** Benjamin Fritz, Gábor Horváth, Ruben Hünig, Ádám Egri, Markus Guttmann, Marc Schneider, Uli Lemmer, Guillaume Gomard.

**Visualization:** Benjamin Fritz, Gábor Horváth, Ádám Pereszlényi, Guillaume Gomard.

**Writing – original draft:** Benjamin Fritz, Gábor Horváth, Ruben Hünig, Marc Schneider, Uli Lemmer, György Kriska, Guillaume Gomard.

**Writing – review & editing:** Benjamin Fritz, Gábor Horváth, Ruben Hünig, Uli Lemmer, Guillaume Gomard.

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
