## [Decision Letter · Decision Letter 0]

3 Nov 2020

PONE-D-20-26794

Multifunctional Rose-Petal-Mimicking Light-Harvesting Layers for Solar Panels Nearly Eliminate Polarised Light Pollution

PLOS ONE

Dear Dr. Horvath,

Thank you for submitting your manuscript to PLOS ONE. After careful consideration, we feel that it has merit but does not fully meet PLOS ONE’s publication criteria as it currently stands. Therefore, we invite you to submit a revised version of the manuscript that addresses the points raised during the review process.

In particular:

1) Please improve the readability of the introduction, per comment from Reviewer 1;

2) Please add supporting data, per comments from all three Reviewers, if possible, to either the main text or the supporting information. If it is not possible, please explain why it was not possible to add the requested data in your Response to the Reviewers.

We look forward to receiving your revised manuscript.

Kind regards,

Oksana Ostroverkhova

Academic Editor

PLOS ONE

Journal Requirements:

Reviewers' comments:

Reviewer's Responses to Questions

**Comments to the Author**

1. Is the manuscript technically sound, and do the data support the conclusions?

Reviewer #1: Yes

Reviewer #2: Yes

Reviewer #3: Yes

2. Has the statistical analysis been performed appropriately and rigorously? 

Reviewer #1: Yes

Reviewer #2: Yes

Reviewer #3: Yes

3. Have the authors made all data underlying the findings in their manuscript fully available?

Reviewer #1: Yes

Reviewer #2: Yes

Reviewer #3: No

4. Is the manuscript presented in an intelligible fashion and written in standard English?

Reviewer #1: No

Reviewer #2: Yes

Reviewer #3: Yes

5. Review Comments to the Author

Reviewer #1: The authors of this study devise a way to reduce polarized light pollution from solar panels by using rose petaled shapes of material, which break up the polarizing effect of solar panels. They measured the polarization of the solar panels extensively and included two behavioral tests with insects: mayflies and horseflies. They found that mayflies and horseflies were less likely to fly to the rose petaled panel. Overall the science is very strong and the impact of this study is very enlightening for both basic (visual ecology and photovoltaic science) and applied (conservation and solar panel development) sciences. However, I do have some reservations and constructive criticisms before the manuscript be fully accepted for publication.

The English and overall writing of the manuscript is quite good, however, I had a very difficult time getting through the manuscript due to all of the abbreviations and jargon. For a manuscript in PLOS, the writing style should be accessible to the emerging scientist (i.e. undergraduate student) and non-scientist (preferably at a high school level) and unfortunately this paper does not meet that requirement. My partner, whom is a journalist, couldn't get through the first paragraph without looking stuff up or asking very specific questions. So, please rewrite the introduction - at least the first few paragraphs - to be more accessible to PLOS readers. Also, get rid of most of the abbreviations. I don't understand why you have an abbreviation for polarized light pollution or rose petal - just write it out. The abbreviation comes across as lazy and does not help with the manuscript. I would suggest getting rid of all of the abbreviations except for the chemical and UV, Vis abbreviations.

I think you should use solar panel somewhere in your title or keywords so that it comes up if people don't search photovoltaic.

The figures are well done but I am wondering if there is a way to add to figure 3 to show the amount of polarization? Could you add the mean polarization in figure 3B for each panel? It's difficult to compare between the 5 panels in each photo and to tell the actually difference in polarization.

In the discussion it would be good to discuss a little more about the experimental implications of the mayflies and horseflies. I am interested in hearing the authors thoughts on whether these will greatly reduce attraction if the rose petaled panels are by themselves and the insects don't have surfaces that are more polarized? It makes sense that the insects will go to the more polarized surface, which could be the case if the only surface is the rose petal surface. Please discuss this more in each of the two insect sections in section 3.

Overall, I really liked the study and the ingenious behind it all. I was frustrated by the abbreviations and how it was written for people only with a PhD in optics or visual sciences. I think a more welcoming first couple of paragraphs in the introduction will drastically drive up the number of people who read the manuscript.

Reviewer #2: I congratulate the authors on a rigorously designed, executed, documented study, described in an easy to follow fashion in excellent English. I only have a few minor comments.

1. the title is very technical and does not advertise the biological aspect of the article. I suggest to modify e.g. to "Nanostructured coatings for solar panels nearly eliminate light pollution that harms polarotactic insects" or "Biomimetic nanostructured coatings..."

2. L426 and elsewhere - you talk about the DoP threshold of polarotactic aquatic insects, could you please specify some numbers or at least provide citations?

3. The difference between DoP of surfaces, imaged during horsefly and mayfly experiments, is quite striking. I had to wait until the tables in the Supplement to get an explanation. Mayflies, who experienced DoP >90%, were counted at the sunset, horseflies at around noon. At sunset the skylight is strongly polarized and it was the only illuminant of the panels. So DoP of reflections must be a consequence of complex interactions between the object properties and natural illuminant. If you agree, then please clarify this in the text.

4. The artificial surfaces are always compared to water surfaces. Although it may seem trivial, I regret that water was not imaged at the time of the experiments - due to the above mentioned complex interactions with the illuminant. Of course there are so many types of water, and these measurements should be repeated on the same place around the same time in the year and the day, but I really do not require further experiments. I am sure water has been measured and modeled many times, so please try to be quantitative and specify the values for DoP/AoP of water.

Reviewer #3: In their manuscript entitled „Multifunctional Rose-Petal-Mimicking Light-Harvesting Layers for Solar Panels Nearly Eliminate Polarised Light Pollution“, Fritz et al present new ‘textured photovoltaic cover layers’ for reducing ‘polarized light pollution’ (PLP) emanating from solar panels. This PLP can affect the behavior of many water-seeking insects (shown here by counting the effect of different surfaces on Mayflies and Horse flies), and effectively trap them (or at least lead them to erroneously lay eggs on these surfaces).

The data presented here is very convincing and leaves little room for criticism. I have listed some minor points below. Otherwise I recommend this article for publication in PLoS One.

Minor comments:

Lines 58/59: “…were almost unattractive to these species, and thus greatly reduced PLP”. The logic of this sentence seems wrong, or the statement exaggerated, in my opinion. It should be: “…were almost unattractive to these species, which is indicative of reduced PLP”.

Lines 223/224: Why was transmission not measured? Does the data exist?

Lines 310-312: Polarimetric images of the asphalt would have helped making that claim.

Lines 326 and 335/336: It’s a bit weak that one has to trust an ‘experienced observer’, instead of having photographic/movie evidence. In the future, I recommend documenting these observations, in order to reduce personal bias.

Lines 364-366: Referring to unpublished and unprovided data is weak. One sentence explaining this claim would be helpful.

6. PLOS authors have the option to publish the peer review history of their article (what does this mean?). If published, this will include your full peer review and any attached files.

Reviewer #1: No

Reviewer #2: No

Reviewer #3: No

---

## [Author Response · Author response to Decision Letter 0]

17 Nov 2020

The response to Reviewers is uploaded separately.

---

## [Editor Report · Decision Letter 1]

19 Nov 2020

Bioreplicated coatings for photovoltaic solar panels nearly eliminate light pollution that harms polarotactic insects

PONE-D-20-26794R1

Dear Dr. Horvath,

We’re pleased to inform you that your manuscript has been judged scientifically suitable for publication and will be formally accepted for publication once it meets all outstanding technical requirements.

Kind regards,

Oksana Ostroverkhova

Academic Editor

PLOS ONE
---

## [Editor Report · Acceptance letter]

23 Nov 2020

PONE-D-20-26794R1 

Bioreplicated coatings for photovoltaic solar panels nearly eliminate light pollution that harms polarotactic insects 

Dear Dr. Horvath:

I'm pleased to inform you that your manuscript has been deemed suitable for publication in PLOS ONE. Congratulations! Your manuscript is now with our production department. 

Kind regards, 

on behalf of

Prof. Oksana Ostroverkhova 

Academic Editor

PLOS ONE